# Study on Smoke Leakage Performance of Suspended Ceiling System

**Tien-Lun Chou [1], Chieh-Hsin Tang [2], Ying-Ji Chuang [1],* and Ching-Yuan Lin [1]**

[1]  Department of Architecture, National Taiwan University of Science and Technology, Taipei 10607, Taiwan; choubig8@gmail.com (T.-L.C.); linyuan@mail.ntust.edu.tw (C.-Y.L.)

[2]  Department of Architecture, China University of Technology, Taipei 11695, Taiwan; tang3307@gmail.com

\*  Correspondence: d9413005@gmail.com; Tel.: +886-2-2737-0259; Fax: +886-2-2737-0538

**Abstract:** The key focus of the research is on the smoke leakage rate from suspended ceiling system, referencing CNS 15038 norm and its experimental principles to build a set of monitoring equipment for measuring air leakage rate and the provision of detailed assembly details for users' reference. Through the real-size test chamber, the smoke insulation performance of the ceiling is studied. Targeting the different ceiling materials, ceiling panels dimensions, and construction methods, in keeping with the scientific principles of fluid mechanics, a total of 405 tests are carried out to come up with the means of appraising the leakage rate of ceiling panels of different sizes and materials. The study found that with the ceiling panel material quality being different, even if the ceiling size is the same, different leakage rates could occur. When the material quality of the ceiling panels is the same and the ceiling size is different, it is not that the larger the size of the panel, the greater the leakage rate, but the smallest leakage rate is caused by the largest panel and this is a very special phenomenon. This study also presents a leakage rate assessment table for entire ceiling panels, which will provide future calculations of the smoke leakage rate of the non-flame room, which can be extrapolated to assess the time of smoke decline and conducive for evacuation design. The apparatus has been proven to have proper leakage rate detection capability for the ceiling panels. In the future, the design principle of the extended system can be applied to the inspection and testing of smoke insulation capability of other fire prevention products. In turn, it can be estimated when the smoke has fallen to facilitate escape design.

**Keywords:** smoke leakage performance; suspended ceiling system; pressure difference

## 1. Introduction

When a fire occurs, the products and airborne particulates produced by the contents of the room will have a noticeable effect on the human respiratory organs. Smoke production in a fire is dangerous to occupants and will cause panic among evacuees and a stimulus-response effect for the evacuation personnel. In general, smoke tends to reach an unbearable level earlier than the temperature [1]. For example, a fire occurred in a building in the early hours of 13 August 2018, at the Taipei Hospital Nursing Home in Taiwan, resulting in a total of 14 deaths and more than 30 people injured. As smoke spreads toward the aisles and other open rooms, as well as due to low relative air density and higher temperatures than usual, it causes the air circulation to move upwards, spreading quickly to the bedrooms and even diffusing towards the entire floor space. As a result of the exposure of people inside the building to smoke, the goal of reducing the number of casualties can be achieved by confining the smoke generated during combustion to a certain area. However, the interior of the building requires many openings for natural lighting, ventilation, access needs or opening sections related to pipelines for daily needs, including doors, windows, elevator openings, or pipelines maintenance

doors, which are one of the routes in which smoke is transmitted from the high-tension side to the step-down side [2] and therefore require special attention to the openings of these buildings.

At present, countries have drawn up testing norms for the air leakage volume of doors, such as CNS 15038 [3], ISO 5925-1 [4], ISO 5925-2 [5], JIS A1516 [6], BS 476-31 [7], DIN 18095-1 [8], DIN 18095-2 [9], UL 1784 [10], and ASTM E 283 [11]; Taiwan's test standard is the CNS 15038 [3], which is in line with ISO 5925-1 [4].The test method is the same. The test content of CNS 15038 [3] is mainly divided into two parts: (1) The specimen shall pass through a volume leakage of not more than 25 $m^3$/h at normal temperature and at a differential pressure of 25 Pa; (2) the specimen shall pass a volume leakage of not more than 25 $m^3$/h at mesothermal condition (200 °C ± 20 °C) and 25 Pa differential pressure. The requirements of CNS 15038 [3] are only met when the doors are below the volume leakage rates specified in both parts above. The smoke-control performance of doors has been examined through correlation studies by several people, like Chuang et al. [12] who pointed out that the doors are frequently opened and closed, the volume of air leakage between the "old door" of a building and the "new door" of the laboratory will vary, which will also cause wear and tear to the door fittings over time. Kuo [13] proposed a set of methods for testing smoke-shielding doors on-site to understand the smoke insulation performance of the doors installed on the construction site. Chuang et al. [14] also suggested that the leakage of the lower door gap accounted for about 88% of the entire door leakage. Even Lin et al. [15] tested the smoke shielding performance of the landing door, and Tsai et al. [16] tested the smoke shielding performance of fire-stopping, but for a safe room. In addition to paying attention to the leakage of doors and fire stops, the space above the ceiling panels may also be one of the paths for smoke transmission. Klote [17] has conducted an actual outbreak of fire tests in the hospital wards and used the air supply and exhaust system to transmit the smoke from the fire room to the non-fire sites, to measure the carbon monoxide content and the smoke density. Although there is a reference to the suspended ceiling system, it is only limited to the discussion of the smoke movement. However, there is no relevant research on the details of the ceiling panels construction method and the leakage rate. In other words, the quality and gap of the suspended ceiling system have a bearing on the smoke spread velocity between the rooms. If we can understand the details of this part, we should be able to improve the overall safety of the living room.

Since the doors are vertical and the ceilings are horizontal structures, a comprehensive survey of all the world's test specifications [3–11] reveals they are for smoke insulation differential pressure tests only. Therefore, the study hopes to use a new set of air leakage detector, modeled on the requirements of norm CNS15038 [3] test methods and scientific principles to measure the smoke insulation performance of the ceiling to enhance the safety of the building. Besides, since the test specifications [3–11] of various countries only set test standards, which are simply stated in the specifications, it takes quite a while for the test personnel to set up the instrument, the study will provide detailed information and principles of the test equipment unreservedly to subsequent relevant personnel for reference and use, and related analysis of the smoke insulation performance of the suspended ceiling system is proposed. In this study, the commonly used thick gypsum board, glass-fiber board, and calcium silicate board are used as samples in a test without regard to the material effects of other relevant lighting fixtures and return air boards to focus on the smoke leakage rate and material quality of the ceiling panels of the suspended ceiling system so that in future, buildings planning personnel can use them as a reference basis for the design.

## 2. Experimental Plan

### 2.1. Experimental Apparatus

The test equipment is based on the same equipment principles as used by Kuo [13] and is built to the CNS 15038 [3] norm, which allows for the measurement of the leakage of the specimens at room temperature. As shown in Figure 1, it can be broadly divided into three main parts, with the first part

being the air blower, the second part being the flow meter, and the third part being the test chamber (install manometer and specimen).

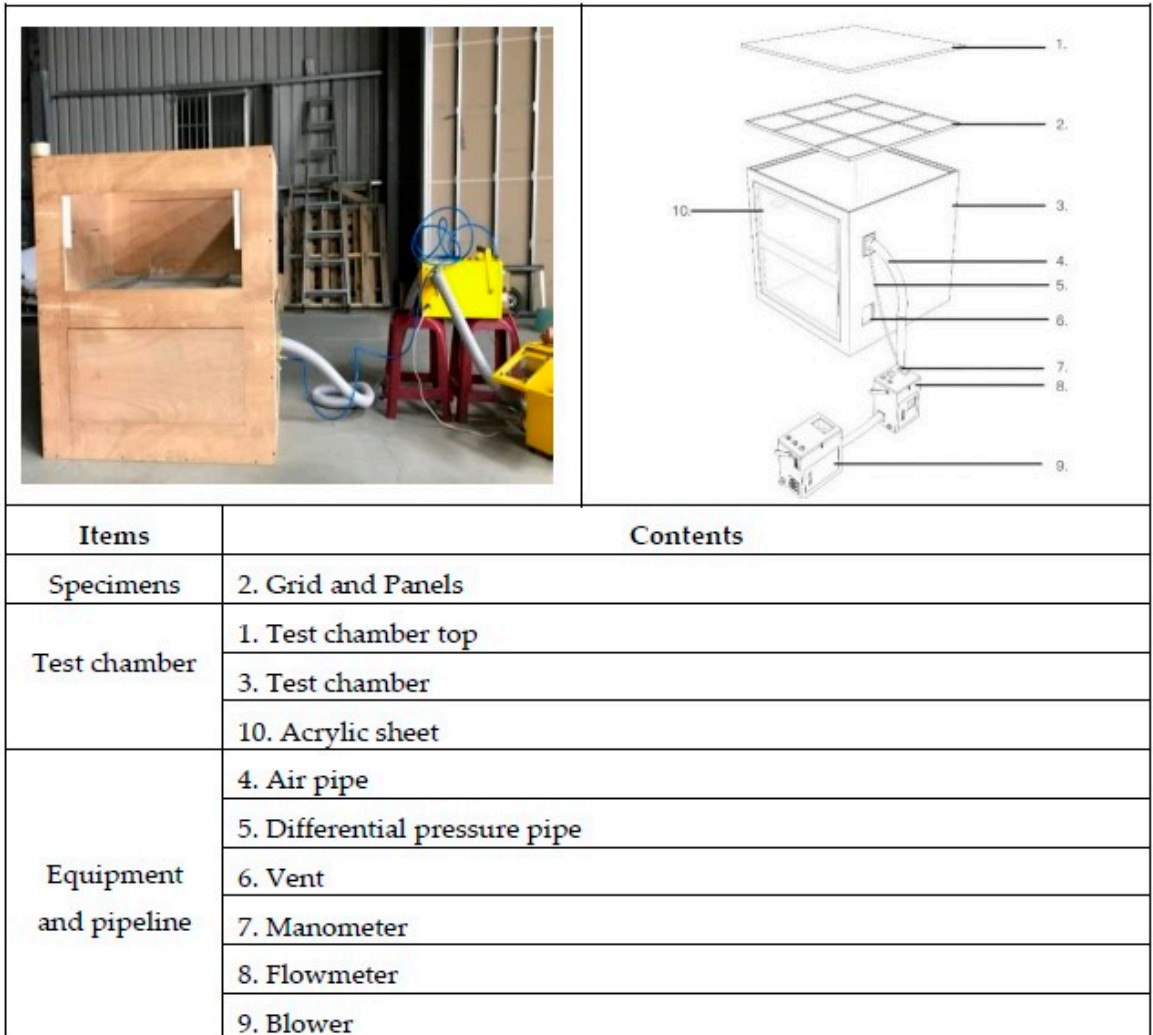

| Items | Contents |
|---|---|
| Specimens | 2. Grid and Panels |
| Test chamber | 1. Test chamber top |
| | 3. Test chamber |
| | 10. Acrylic sheet |
| Equipment and pipeline | 4. Air pipe |
| | 5. Differential pressure pipe |
| | 6. Vent |
| | 7. Manometer |
| | 8. Flowmeter |
| | 9. Blower |

**Figure 1.** Experimental apparatus.

The first part is the blower from Y.H. Industrial Co., Ltd. (Taipei, Taiwan), the maximum air volume is 6.8 m$^3$/min, it has $\frac{1}{4}$ HP, uses 220 V, three-phase electricity, and has a 50 mm air outlet diameter. Combined with a Teco Electric and Machinery Co., Ltd. (Taipei, Taiwan) variable frequency drive to control the blower's rotation speed (frequency control range 0.01~650.00 Hz).

The second part is the flowmeter, a Honeywell (Honeywell International, Inc., Charlotte, NC, USA) smart differential pressure transmitter used with a flowmeter. The flowmeter can measure from 0 to 75 m$^3$/h with an accuracy of ±2.5%. It is suitable for use with fluid temperature between −10–60 °C and the humidity is below 90%. The flowmeter is located between the blower outlet and the test chamber. The outlet and inlet pipe diameter of the flow meter is 50 mm.

The test chamber structure of the third part is composed of a lumber core board of 24 mm in thickness and a density of 0.353 g/cm$^3$ and is provided with a transparent acrylic plate (for observing the change in the specimen). The inner dimensions of the test chamber are 1225 × 1225 × 1225 mm, and the joint of the wooden core board is glued with a strong adhesive. The joints of the inner wood core boards of the test chamber are adhered to the surface with airtight tape. The test chamber has a hole on the side and is fitted with a Testo 510 (Testo SE and Co. KGaA, Titisee-Neustadt, Germany)

pocket-size differential pressure gauge having a measuring range of 0 to 100 hPa with an accuracy of ±0.03 hPa.

The fourth part is a thermometer, a relative hygrometer, and a barometer. The measuring range of the thermometer is −40 °C to 100 °C, and the resolution is 0.1 °C. The measuring range of the relative hygrometer is 0% RH to 100% RH with a resolution of 0.1% RH. The barometer has a measuring range of 300 to 1200 hPa and a resolution of 0.1 hPa.

## 2.2. Experimental Samples

Hospitals generally use suspended light steel frame ceilings, which are easier to work on than traditional woodwork ceilings and which can be removed and maintained to facilitate pipeline maintenance. The study uses the usual thickness 9 mm gypsum board, 15 mm glass-fiber board, and 3.5 mm calcium silicate board samples with densities of 0.75 g/cm³, 0.064 g/cm³, and 1.05 g/cm³, respectively. The load bending rupture being 36.3 kg/cm², 5.1 kg/cm², and 13.1 kg/cm², respectively. The dimensions of the three panels are 603 × 603 mm (2′ × 2′). The size and type of the exposed ceiling grid are shown in Figure 2.

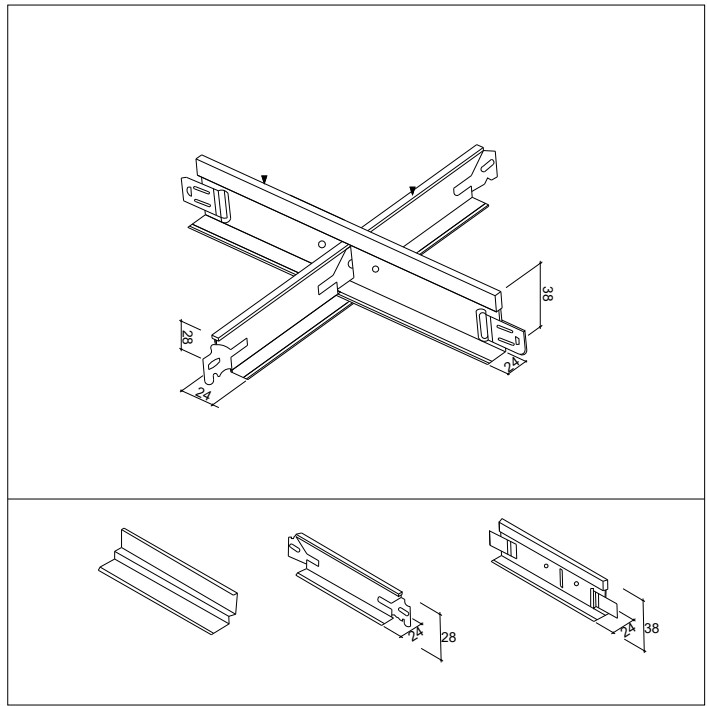

**Figure 2.** The size and type of ceiling grid (unit: mm).

## 2.3. Experimental Procedure

This procedure is carried out under CNS 15038 [3].

First: Determine if the leakage volume of the test chamber itself meets the requirements of norm CNS 15038 [3].

The airtight tape is first applied to the crack of the test chamber. When the differential pressure is 55 ± 1 Pa, the leakage rate is corrected to the gas standard condition (temperature 20 °C (293.15 K) and standard 1 atmospheric pressure (101,325 Pa)). It shall not exceed 7 m³/h to meet the basic leakage requirements of CNS 15038 [3] test chamber.

CNS 15038 [3] requires the actual measured leakage rate to be corrected to the post gas standard condition algorithmic method of gas calculation Equation (1) as follows:

$$Q_a' = \frac{Q_a}{(T + 273.15)} \times \left[ k \times (p_a + p_m) - 3.795 \times 10^{-3} \times M_w \times P_{H2O} \right] \qquad (1)$$

$Q_a'$ Actual leakage of the test specimens under gas standard condition (m³/h);
$Q_a$ Actual leakage of the test specimen at temperature ($T + 273.15$) and pressure ($P_a + P_m$) (m³/h);
$Q_b$ Leakage of the test chamber itself (m³/h);
$Q_t$ Leakage of the test specimen and the test chamber itself (m³/h)
$k$ Constant ($2.89 \times 10^{-3}$);
$T$ Air temperature (°C);
$P_a$ Atmospheric pressure (Pa);
$P_m$ Pressure added-value (Pa);
$M_w$ Relative humidity (%);
$P_{H2O}$ Saturation vapor pressure (Pa);
Second: Leakage measurement of specimens.

After loading the specimen, the test is also carried out at this stage following norm CNS 15038 [3], and leakage measurements of the specimens at 10, and 50 Pa differential pressures are carried out.

Step 1: Measure the leakage $Q_b$ of the test chamber itself.

Step 2: After the test specimen is loaded, measure the leakage $Q_t$ between the test specimen and the test chamber itself.

Step 3: The actual leakage of the test specimen is obtained by subtracting the value in Step 2 from the value in Step 1:

$$Q_a = Q_t - Q_b \qquad (2)$$

Step 4: Convert the value obtained from step 3 $Q_a$ to the leakage volume $Q_a'$ at the gas standard condition (temperature 20 °C (293.15 K) and standard 1 atmospheric pressure (101,325 Pa)).

## 3. Results and Discussion

### 3.1. Basic Leakage of Apparatus

First determine if the leakage volume of the test chamber itself complies with the norm CNS 15038 [3] according to the method of Section 2.3. Because the lumber core board of the test chamber structure is air-locked, the seam of the wooden core board is glued with wood-based strong adhesive. In addition, the junction of the inner core board of the test chamber is adhered to the surface with airtight tape, which can prevent the tape from being blown away by wind pressure. It can completely ensure the air tightness and durability of the crack. First, slowly increase the air flow of the blower so that the differential pressure in the test chamber reaches 55 ± 1 Pa and continuously observe for 10 min. The differential pressure is stable and can be controlled at 55 Pa, indicating that the test chamber and the pipeline are very airtight and will not increase the leakage with time increase. At this point, the flowrate of the flow meter is recorded and displays as 0.4 m³/h, the temperature of the day is measured at 28.0 °C, the relative humidity is 77%, atmospheric pressure is 101,220 Pa, and saturation water vapor pressure is 3781.8 Pa (saturation water vapor pressure is based on the 1946 Goff–Gratch [18] equation, Equation (2); the saturation vapor pressure of the water can be determined at a specified temperature). The basic leakage volume of the test chamber is 0.4 m³/h and according to CNS 15038 [3] norm, the measured leakage volume is required to be revised to 0.37 m³/h, which is less than the gas standard condition (at 20 °C (293.15 K) and with atmospheric pressure of Standard 1 (101,325 Pa)) requirement of 7 m³/h.

Therefore, the basic leakage volume of the test chamber is in compliance with the requirements and is feasible as a follow through test specimen measurement. The basic leakage measurement of this test chamber shall be carried out once before the test on the same day, and therefore the basic leakage measurement of each test chamber will not be similar. This test chamber combination, tape adhesive method, and revision are calculated for the gas standard condition, which can provide a reference for those who wish to conduct related tests in the future. Of course, some may think of the significance of the requirement that the test module shall not exceed 7 m$^3$/h after the test module has been modified to a gas standard condition (at temperature 20 °C (293.15 K) and a standard 1 atmospheric pressure (101,325 Pa)). After the test specimen has been loaded, the leakage of the specimen and the test chamber itself is then deducted. Deducting again the leakage of the test chamber itself, the actual leakage of the specimen value is for comprehension, the size of the value is not related to the actual leakage measure $Q_a$ which is actually correct in theory. The calculation of the actual leakage rate of the specimen by countries is also the total leakage minus the basic leakage of the test chamber [3–11].

But if we take the idea of "air pressure leak" [19], it is the most often neglected phenomenon. When leakage rate occurs, the effect produced can cause a small number of problems, such as increased electricity costs, pressure fluctuations, and reduced blower service life; for the long-term users, these problems that occur are still hopefully avoided. Other regulations concerning leakage levels in test chamber vary from country to country, but the allowable leakage rate of test chambers is small [3–11]; for example, BS476-31 [7] under pressure differential 50 Pa, the corrected leakage of test chambers shall not exceed 7 m$^3$/h; for norm CNS 15038 [3], under pressure differential 55 Pa, the leakage volume of the test chamber, after correction, must not exceed 7 m$^3$/h. In CNS 15038 [3], the specification requires that the differential pressure be not greater than 25 m$^3$/h after the leakage correction of 25 Pa, and that if the leakage of the test chamber itself does not specify a maximum value, such as when the modified leakage of the test chamber itself is greater than 25 m$^3$/h, the corrected leakage of the test body is also close to 25 m$^3$/h, total corrected leakage above 50 m$^3$/h may cause excessive leakage, preventing the differential pressure gauge and flow meter from reading the value steadily, resulting in a failure to perform the test smoothly, so this study is compared to CNS 15038 [3]. It is reasonable to require that the leakage volume of the test chamber itself, after correction, not exceed 7 m$^3$/h. It also allows the test process to be simple and oriented toward the proximity of the flow of the current and steady state.

The Goff–Gratch [18] Equation (3) is as follows:

$$
\begin{aligned}
\log e^x = \ & -7.90298\left(\tfrac{T_{st}}{T}-1\right)+5.02808\log\left(\tfrac{T_{st}}{T}\right)-1.3816 \\
& \times 10^{-7}\left(10^{11.344\left(1-\tfrac{T}{T_{st}}\right)}-1\right)+8.1328 \\
& \times 10^{-3}\left(10^{-3.49149\left(\tfrac{T_{st}}{T}-1\right)}-1\right)+\log e^*_{st}
\end{aligned}
\tag{3}
$$

in which:

log: Logarithm to base 10
$e^*$: Saturated vapor pressure of water (hPa)
$T$: Absolute temperature in Kelvin (K)
$T_{st}$: Boiling point temperature (373.15 K)
$e^*_{st}$: Boiling point pressure (1 atm = 1013.25 hPa)

### 3.2. Leakage of Each Ceiling Material

Nine-millimeter-thick gypsum board, 15 mm-thick glass-fiber board, and 3.5 mm-thick calcium silicate board are used as samples. The standard size of each ceiling panel is 603 × 603 mm (2′ × 2′). Since the production of the ceiling panels is fully automatic machine production with standardized dimensions and grids, the panels ensure flatness and speed during construction, hence the advantage. The ceiling grid can be modified with fine tools, pliers, and shears on site. The internal width and

length of the test chamber are 1.225 m, so the test chamber can accommodate a complete ceiling width of 603 mm × length 603 mm, four pieces width 300 mm × length 603 mm, and four pieces of 300 mm × length 300 mm ceilings, as shown in Figure 3. This ceiling plan can be divided in a way that echoes the actual installation, because it is not possible to lay all the pieces of the ceiling in practice, and there must be sections that need to be cut near the edge of the wall. Therefore, the test chamber can be planned for two purposes; one to conform to real situation, and the other, to be used daily for different sizes of boards, can be used to analyze leakage rate in ceilings of various sizes. Since the actual measurement of leakage rate required by CNS 15038 [3] in this study needs to be revised to the gas standard condition for comparison, to facilitate subsequent discussions the term "leakage rate" appears in the following to deduct the leakage from the basic test chamber. The leakage rate is the leakage rate corrected to the gas standard condition, in other words, it is obtained by applying Equation (1) the result $Q_a'$ (the actual leakage rate of the sample in the gas standard condition ($m^3/h$). The three materials are applied to the ceiling decoration and are generally laid using only a single material each, with little mixing and matching. Therefore, each test is done using a single material, and the ceiling is restored after each test. The test is done for a total of five consecutive times and the average leakage rate is obtained objectively and randomly. Number the ceiling panels and test sequentially. First, use the knob of the blower to slowly adjust the differential pressure to 10 Pa, and record the corresponding leakage rate, and then slowly adjust the knob of the blower to 25 Pa, and record its corresponding leakage rate.

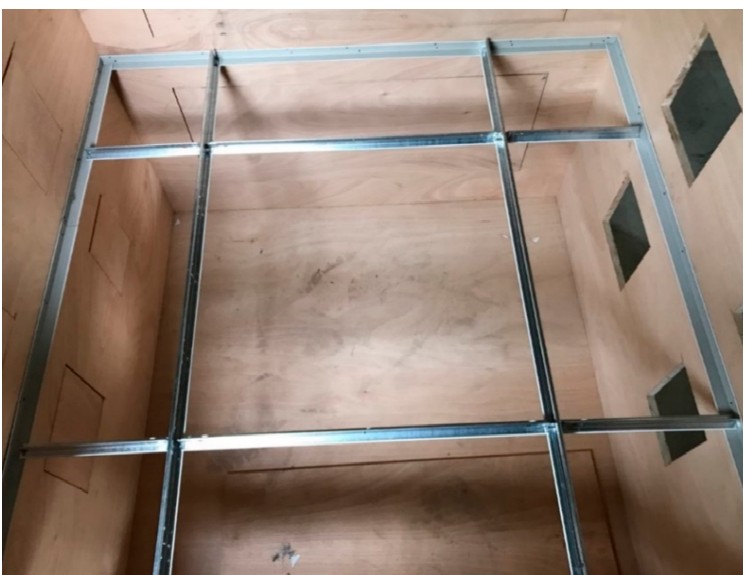

**Figure 3.** The test chamber.

Type 1: Test 9 mm gypsum board, as shown in Figure 4, apply adhesive tape to ceiling crack outside ceiling panel no. 5 to obtain leakage of ceiling no. 5 (No. A5). The average leakage volume of A1, A2, A3, A4, A6, A7, A8, and A9. A10 being the grand average leakage volume from all the ceiling panels.

Type 2: Test 15 mm-thick glass-fiber panels, taking the average leakage from each ceiling panel according to sequence, to obtain the average leakage from each ceiling panel number B1, B2, B3, B4, B5, B6, B7, B8, and B9. B10 being the grand average leakage volume from all the ceiling panels.

Type 3: Test the 3.5 mm-thick calcium silicate board, taking the average leakage from each ceiling panel according to sequence, to obtain the average leakage from each ceiling panel number C1, C2, C3, C4, C5, C6, C7, C8, and C9. C10 is the total average leakage volume from all the ceiling panels.

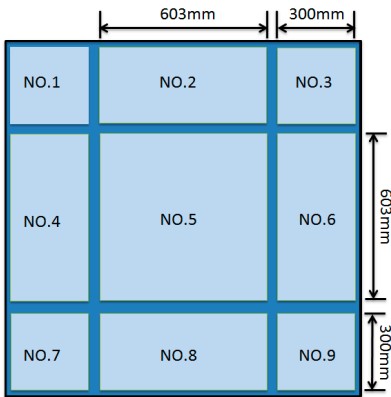

**Figure 4.** Schematic diagram of ceiling numbering.

The leakage of each ceiling panel is obtained by averaging five times and a total of 405 tests is done. The results of which are shown in Table 1.

**Table 1.** Average leakage of ceiling material at each pressure difference (m$^3$/h).

| Specimen Pressure Difference | 9 mm Gypsum Board Number A | | 15 mm Glass-Fiber Board Number B | | 3.5 mm Calcium Silicate Board Number C | |
|---|---|---|---|---|---|---|
| | A1 | 11.01 | B1 | 1.89 | C1 | 10.11 |
| | A2 | 17.92 | B2 | 2.52 | C2 | 16.02 |
| | A3 | 11.10 | B3 | 1.90 | C3 | 10.01 |
| | A4 | 17.95 | B4 | 2.48 | C4 | 15.98 |
| | A5 | 8.54 | B5 | 1.72 | C5 | 6.09 |
| 10 Pa | A6 | 17.85 | B6 | 2.61 | C6 | 16.00 |
| | A7 | 11.15 | B7 | 1.99 | C7 | 10.12 |
| | A8 | 17.86 | B8 | 2.50 | C8 | 15.99 |
| | A9 | 11.21 | B9 | 1.92 | C9 | 10.12 |
| | (A1 + A3 + A7 + A9)/4 | 11.12 | (B1 + B3 + B7 + B9)/4 | 1.93 | (C1 + C3 + C7 + C9)/4 | 10.09 |
| | (A2 + A4 + A6 + A6)/4 | 17.90 | (B2 + B4 + B6 + B6)/4 | 2.53 | (C2 + C4 + C6 + C6)/4 | 16.00 |
| | A10 | 124.59 | B10 | 19.53 | C10 | 110.44 |
| | A1 | 19.44 | B1 | 3.42 | C1 | 17.56 |
| | A2 | 23.08 | B2 | 3.97 | C2 | 22.92 |
| | A3 | 19.50 | B3 | 3.33 | C3 | 17.25 |
| | A4 | 23.12 | B4 | 3.82 | C4 | 22.88 |
| | A5 | 13.45 | B5 | 2.59 | C5 | 9.67 |
| 25 Pa | A6 | 23.20 | B6 | 3.98 | C6 | 22.98 |
| | A7 | 19.59 | B7 | 3.55 | C7 | 17.45 |
| | A8 | 23.45 | B8 | 3.78 | C8 | 22.78 |
| | A9 | 19.45 | B9 | 3.44 | C9 | 17.79 |
| | (A1 + A3 + A7 + A9)/4 | 19.50 | (B1 + B3 + B7 + B9)/4 | 3.44 | (C1 + C3 + C7 + C9)/4 | 17.51 |
| | (A2 + A4 + A6 + A6)/4 | 23.21 | (B2 + B4 + B6 + B6)/4 | 3.89 | (C2 + C4 + C6 + C6)/4 | 22.89 |
| | A10 | 184.28 | B10 | 31.88 | C10 | 171.28 |
| | A1 | 27.41 | B1 | 4.98 | C1 | 24.12 |
| | A2 | 36.20 | B2 | 5.97 | C2 | 33.21 |
| | A3 | 27.50 | B3 | 4.97 | C3 | 24.25 |
| | A4 | 36.21 | B4 | 6.25 | C4 | 33.51 |
| | A5 | 16.88 | B5 | 3.51 | C5 | 11.25 |
| 50 Pa | A6 | 36.01 | B6 | 6.15 | C6 | 33.45 |
| | A7 | 27.01 | B7 | 4.78 | C7 | 24.22 |
| | A8 | 36.09 | B8 | 5.88 | C8 | 33.12 |
| | A9 | 27.11 | B9 | 4.75 | C9 | 24.01 |
| | (A1 + A3 + A7 + A9)/4 | 27.26 | (B1 + B3 + B7 + B9)/4 | 4.87 | (C1 + C3 + C7 + C9)/4 | 24.15 |
| | (A2 + A4 + A6 + A6)/4 | 36.13 | (B2 + B4 + B6 + B6)/4 | 6.06 | (C2 + C4 + C6 + C6)/4 | 33.32 |
| | A10 | 270.42 | B10 | 47.24 | C10 | 241.14 |

## *3.3. Analysis and Application*

As shown in Table 1, the maximum leakage is found on panels materials A, the minimum leakage is on the B panels, regardless of the differential pressure of 10 Pa, 25 Pa, and 50 Pa. The leakage at 10 Pa in sequence of A10 (132.59 m$^3$/h) in order. C10 (110.44 m$^3$/h) > B10 (19.53 m$^3$/h), the leakage at 25 Pa in sequence of A10 (184.28 m$^3$/h) > C10 (110.44 m$^3$/h) > B10 (31.88 m$^3$/h). The leakage rate at 50 Pa

in sequence of A10 (270.42 m$^3$/h) > C10 (241.14 m$^3$/h) > B10 (47.24 m$^3$/h). The larger the differential pressure, the greater the total leakage, the more regular the test results (see Figure 5). However, it is very strange that since the lengths and widths of the three types of panels are the same, in theory, the coverage area placed on the grid is exactly the same, and it is logical that they should have the same leakage. This result overturns every man's cognition and analyzes the cause of it, which can be explained by the physical nature of the panels themselves. Since the load bending rupture of panel A is 36.3 kg/cm$^2$, it is greater than the load bending rupture of panel C13.1 kg/cm$^2$, and the load bending rupture of panel B is 5.1 kg/cm$^2$. When the wind exerts pressure from above the ceiling, the ceiling material with relatively low flexural strength tends to compact with the crack, which makes the airflow unable to pass through the crack. Since the main component of the gypsum board is gypsum, the special paper is used to wrap the gypsum. The density is lower than that of the calcium silicate board. The goal is to pursue the stability and non-deforming performance of the material. It is just the characteristic of good bending resistance that results in a large amount of leakage.

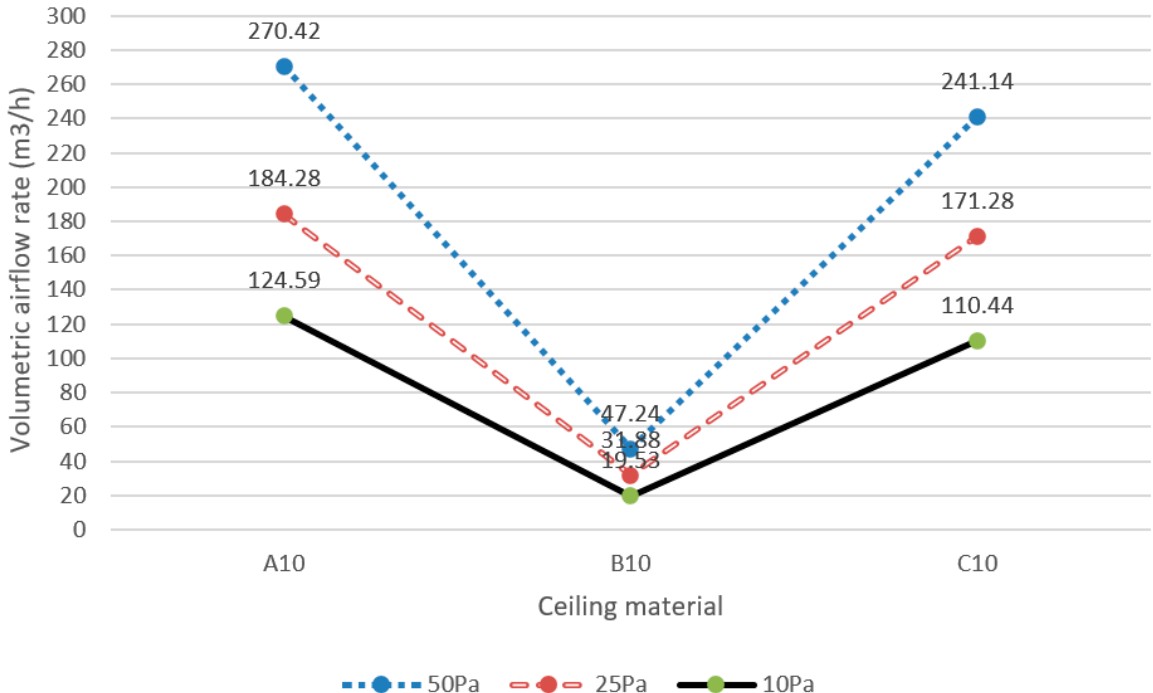

**Figure 5.** Chart of the relationship between ceiling material and leakage at various pressure differences.

The opposite is true; the softest and deformable glass-fiber board unexpectedly has the smallest leakages at 10 Pa, 25 Pa, and 50 Pa pressure difference. This study only focuses on the state of gypsum board in dry conditions, as the CNS 4458 [20] norm specifies that dry boards should have a greater load bending rupture than in moist conditions, with a requirement of 36.0 kg/cm$^2$ or above in dry conditions and a requirement of 14.0 kg/cm$^2$ or more in moist conditions. The moist condition is close to the load bending rupture of 13.1 kg/cm$^2$ of panel C. In theory, when the load bending rupture of the gypsum board becomes smaller after a prolonged period of moisture or other factors, the leakage volume of the gypsum board should theoretically be close to panel C.

Table 1 shows that the size of the panels does affect the leakage rate, which is very close under the same conditions of material, size, and pressure difference. For example, when panels of size of 300 × 300 mm is at a pressure difference of 10 Pa, the leakage of A1, A3, Á7, and A9, is 11.01 m$^3$/h, 11.10 m$^3$/h, 11.15 m$^3$/h, and 11.21 m$^3$/h, respectively, with an average value of 11.1175 m$^3$/h. The standard deviation ($\sigma$) is 0.0733, and the degree of dispersion is low, which demonstrates that the test has ample credibility. Another example, when panels of size 603 × 300 mm are subject to pressure difference of 50 Pa, the leakage volumes of C2, C4, C6, and C8 are 33.21 m$^3$/h, 33.51 m$^3$/h, 33.45 m$^3$/h, and 33.12 m$^3$/h,

respectively, with an average value of 33.3225 m$^3$/h and a standard deviation of 0.1621, the discrete degree is low and demonstrates the credibility of the test. According to Figure 6, it is not that there is a direct relationship between the larger the size of the panel and the larger the leakage, but rather another kind of regularity that is the largest the panel, the smallest the leakage. For example, when A panels are subject to pressure difference of 10 Pa, the leakage volume of 300 × 300 mm panel size is 11.12 m$^3$/h, and the leakage volume of 300 × 603 mm panel size is 17.90 m$^3$/h. The leakage capacity of 603 × 603 mm is 8.54 m$^3$/h. At 25 Pa pressure difference, for the same A panels, the leakage volume of 300 × 300 mm panel size is 19.50 m$^3$/h and the leakage volume of panel size 300 × 603 mm is 23.21 m$^3$/h. The leakage volume of 603 × 603 mm is 13.45 m$^3$/h, which means that the maximum leakage volume is found on 300 × 603 mm panels and the minimum leakage volume on 603 × 603 mm ones. This phenomenon, no matter whether it is Panel A, Panel B, or Panel C, has this regularity and it also overthrows ordinary people's perception, because the largest panel has a perimeter of 1206 mm (= 603 + 603 + 603 + 603), the smallest panel has a perimeter of 600 mm (= 300 + 300). Generally speaking, the panel itself has no crack.

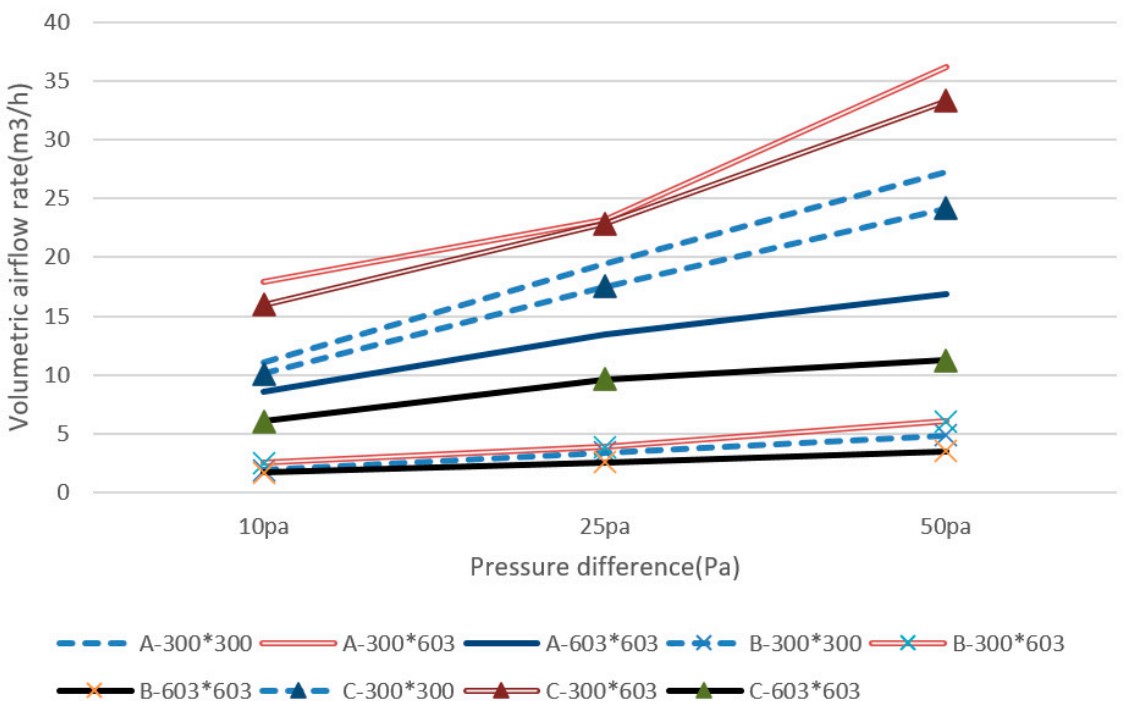

**Figure 6.** Chart of the relationship between the different ceiling sizes, materials, and leakage volumes under each pressure difference.

The more cracks between the grid and the panel, the less airtight it will be. Here, Bernoulli's principle [21] can be used to explain how the air flow behavior can be simplified to analyze the one-dimensional flow assumption:

$$Q = C \times A \sqrt{\frac{2\Delta P}{\rho}} \tag{4}$$

where: $Q$: Volume leakage of air flow through the crack (m$^3$/s)

C: Flowability coefficient
A: Crack area (m$^2$)
$\Delta P$: Differential pressure (Pa)
$\rho$: Air density (kg/m$^3$)

When the flowability coefficient, pressure difference, and air density are the same, while different leakage rates are obtained, then, the only influencing factor is the gap area. When the gap area is larger, the larger the leakage rate, this is constant reasoning. Tracing it to its cause, there are two reasons for this. The first reason is that the size of the panel being 603 × 603 mm, therefore it has a larger deflection, which can put it closer to the main and secondary grids. Therefore, the crack area will be smaller, and therefore the leakage rate will be relatively small. The second reason being that when the panel size is 300 × 603 mm and 300 × 300 mm, it is close to the edge of the wall, therefore it is necessary to use edge-taping material on the wall. However, after cutting, the junctions between the grids and the trim are generally irregular surface, and the junctions have self-tapping screws to fix the panel, and the panel cannot fit the laminating material and the grid, and the crack area is large. Therefore, the leakage of the panel, the size of 300 × 603 mm is greater than that of 300 × 300 mm, as shown in Figure 7.

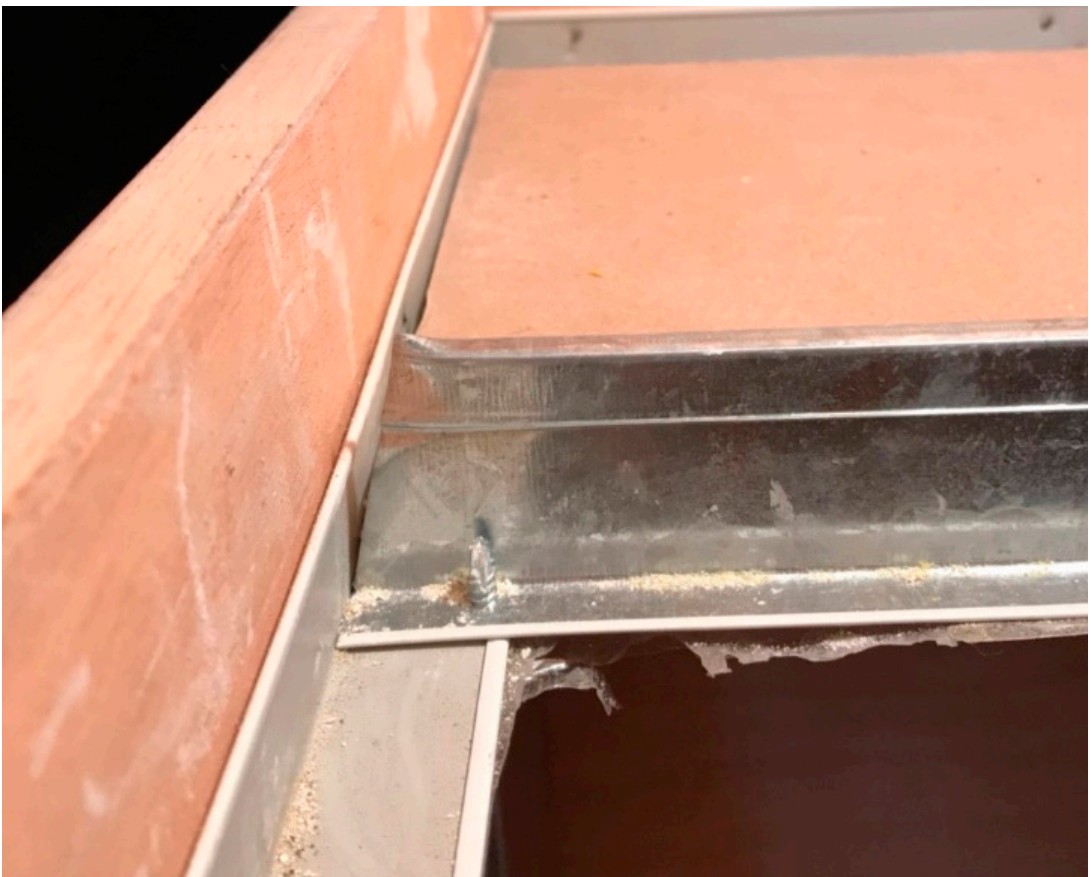

**Figure 7.** Junction of the grid and the trimming material after cutting.

The flatness of panel size, 300 × 603 mm is close to that of a 300 × 300 mm panel, so it can be used to convert the reasonable leakage near the wall. For example, if panel A, size 300 × 603 mm has a leakage rate of 23.21 under 25 Pa m³/h with a perimeter of 1.806 m (= 0.3 + 0.603 + 0.3 + 0.603) and a leakage of 12.85 m³/(h·m) per unit length. The leakage capacity of panel size, 300 × 300 mm is 19.50 m³/h with a perimeter of 1.2 m (= 0.3 + 0.3 + 0.3 + 0.3 + 0.3 + 0.3) and a leakage capacity of 16.25 m³/(h·m) per unit length. The estimated leakage per unit length of the two different size panels is different. For a more reasonable calculation, it is recommended that when the panel size is less than 300 × 300 mm and between 300 × 300 mm and 300 × 603 mm, respectively, it is necessary to substitute it and different inferred estimates are required to obtain more accurate smoke leakage rate per length unit. The ceiling leakage volume is arranged as shown in Table 2. If properly applied, how much pressure difference can be used to evaluate the smoke leakage rate of the non-flame room, and the

smoke leakage rate can be estimated to facilitate evacuation and escape design. It is a standardized and common type, so the results of this study can provide a reference for the relevant national regulations.

**Table 2.** Leakage of ceiling material.

| Specimen Pressure Difference Size | 9 mm Gypsum Board | | | 15 mm Glass-Fiber Board | | | 3.5 mm Calcium Silicate Board | | |
|---|---|---|---|---|---|---|---|---|---|
| | 10 Pa | 25 Pa | 50 Pa | 10 Pa | 25 Pa | 50 Pa | 10 Pa | 25 Pa | 50 Pa |
| Each 603 × 603 (mm) | 8.54 | 13.45 ($m^3$/h) | 16.88 | 1.72 | 2.59 ($m^3$/h) | 3.51 | 6.09 | 9.67 ($m^3$/h) | 11.25 |
| 300 × 300 or less (mm) | 9.27 | 16.25 ($m^3$/(h·m)) | 22.72 | 1.61 | 2.87 ($m^3$/(h·m)) | 4.06 | 8.41 | 14.59 ($m^3$/(h·m)) | 20.13 |
| 300 × 603 ~ 603 × 603 (mm) | 9.91 | 12.85 ($m^3$/(h·m)) | 20.01 | 1.40 | 2.15 ($m^3$/(h·m)) | 3.36 | 8.86 | 12.67 ($m^3$/(h·m)) | 18.45 |

## 4. Conclusions

(1) Different ceiling materials, even with the same ceiling size, produce different leakage rates, which is caused by the varying degrees of bending bursting strength. When the pressure difference is 10 Pa, 25 Pa, and 50 Pa, the maximum leakage is A panel, the minimum leakage is B panel.

(2) When the ceiling material is the same and the ceiling size is different, the leakage volume of the panel size 300 × 603 mm is the largest. It is not the direct relationship of the larger the size of the panel, the greater the leakage, but the smallest leakage volume caused by the largest panel. This phenomenon applies to all three panels, whether A, B, or C, having this regularity.

(3) Recommendations for leakage assessment of the ceiling as a whole can be found in Table 2 when the panel size is less than 300 × 300 mm and between 300 × 300 mm and 300 × 603 mm, substituting different estimation values need to be used to obtain more accurate smoke leakage rate per length, and when the panel size is 603 × 603 mm, the leakage volume can be directly applied.

(4) This study also presents a leakage rate assessment table for entire ceiling panels, which will provide future calculations of the smoke leakage rate of the non-flame room, which can be extrapolated to assess the time of smoke decline and conducive for evacuation design.

(5) The majority of residents who live in places like long-term care institutions are people with disabilities. Considering that mass evacuation of residents may be difficult with limited caregivers, currently these institutions have opted to take shelter in place, supplemented by horizontal evacuation, and lastly vertical evacuation. Therefore, it is recommended that each ward forms a fire safety compartment, protecting residents from the impact of the flame and smoke. Through the study of smoke leakage volume of ceilings, emphasis must be placed on the smoke insulation function of ceilings so as to buy more time when seeking shelter.

**Author Contributions:** Conceptualization, Y.-J.C.; methodology, C.-Y.L.; investigation, C.-H.T.; writing—review and editing, T.-L.C.; All authors have read and agreed to the published version of the manuscript.

**Funding:** The authors thank the National Science Council of the Republic of China for financially supporting this research under Contract No. NSC 100-2221-E-011-129-.

**Conflicts of Interest:** The authors declare no conflict of interest

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
