# Peer review of "Study on Smoke Leakage Performance of Suspended Ceiling System"

_sustainability, doi:10.3390/su12187244_

Round 1
Reviewer 1 Report
A worthwhile study and could lead to some advances in the understanding of suspended ceiling performance in a fire but I have a few questions:
Line 32: Change "non-negligible " to "noticeable"
Line 32-33: Change start of sentence to "Smoke production in a fire is dangerous to occupants and will cause panic......"
Line 38: Change "caused" to "causes"
Line 63: Not sure what is meant by the term "fire stops"? Is this smoke seals you will find in fire doors?
Line 67: Find a more suitable term than "non-flame room".
Line 88 onwards: When designing the test apparatus, were any assumptions made as to how this would represent a real life situation when a fire at ground level would produce hot smoke at approximately 3.5 metres above, where these suspended ceilings would be located. It would be worth discussing any limitations of the apparatus with regard to real life situations.
Line 274: "When the wind exerts pressure from above the ceiling,...". I am confused as to this statement. From experience, the area above a suspended ceiling is usually static up to the actual soffit ceiling, i.e minimal wind pressure variations. Can this be clarified?
Line 334: Figure 7. Some fire resisting suspended ceiling systems use screwed fittings to maintain high levels of fire resistance. Was such a system considered?
Conclusion: Some practical outcomes or suggestions for designing or installing suspended ceiling systems would be useful based upon the results.
Author Response
Dear Editor and Reviewer 1:
Please see the attachment. The paper is corrected and revised according to the reviewers’ reports. Questions are answered one by one, and the revisions are listed to inform their locations in the text.
We are extremely grateful to all the referees’ excellent comments and valuable advices about our paper.
The authors gratefully acknowledge the constructive comments of the Referees.
Sincerely,
Ying-Ji Chuang
Department of Architecture, National Taiwan University of Science and Technology.
#43, Sec.4, Keelung Rd., Taipei, 10607, Taiwan
Tel: +886-2-27370259
Fax: +886-2-27370538
E-mail: d9413005@gmail.com

Reviewer 2 Report
This manuscript "Study on smoke leakage performance of suspended ceiling system", this research shows an original and innovative research developed with appropriate methods and significative results.
Author Response
Dear Editor and Reviewer 2:
Thanks very much for your kind work and consideration on publication of our paper. On behalf of my co-authors, we would like to express our great appreciation to editor and reviewers.
Thank you and best regards.
Yours sincerely,
The authors gratefully acknowledge the constructive comments of the Referees.
Sincerely,
Ying-Ji Chuang
Department of Architecture, National Taiwan University of Science and Technology.
#43, Sec.4, Keelung Rd., Taipei, 10607, Taiwan
Tel: +886-2-27370259

Reviewer 3 Report
The paper is an interesting work, presenting a study on smoke leakage performance of suspended ceiling system. The study is an original work with extensive analyses providing new contributions to the body of knowledge in the related field, and the conclusions derived from the study are also offering some very useful findings. All in all, this is an excellent contribution to knowledge, however, the authors should extend the literature review as well as to improve the conclusion's section, by adding general conclusions from the conducted experimental study and any further study and general recommendations.
Author Response
Dear Editor and Reviewer 3:
Please see the attachment. The paper is corrected and revised according to the reviewers’ reports. Questions are answered one by one, and the revisions are listed to inform their locations in the text.
We are extremely grateful to all the referees’ excellent comments and valuable advices about our paper.
The authors gratefully acknowledge the constructive comments of the Referees.
Sincerely,
Ying-Ji Chuang
Department of Architecture, National Taiwan University of Science and Technology.
#43, Sec.4, Keelung Rd., Taipei, 10607, Taiwan
Tel: +886-2-27370259
Fax: +886-2-27370538
E-mail: d9413005@gmail.com
